# Afadin sorts different retinal neuron types into accurate cellular layers

**Matthew R Lum[1†], Sachin Patel[1†], Hannah K Graham[1†], Mengya Zhao[1], Yujuan Yi[1], Liang Li[2], Melissa Yao[1], Anna La Torre[3], Luca Della Santina[4], Ying Han[1], Yang Hu[2], Derek S Welsbie[5], Xin Duan[1,6]***

[1]Department of Ophthalmology, University of California, San Francisco, San Francisco, United States; [2]Department of Ophthalmology, Stanford University School of Medicine, Palo Alto, United States; [3]Department of Cell Biology and Human Anatomy, School of Medicine, University of California, Davis, Davis, United States; [4]College of Optometry, University of Houston, Houston, United States; [5]Viterbi Family Department of Ophthalmology, University of California, San Diego, San Diego, United States; [6]Department of Physiology and Kavli Institute for Fundamental Neuroscience, University of California, San Francisco, San Francisco, United States

## eLife Assessment

This study demonstrates the critical role of Afadin on the generation and maintenance of complex cellular layers in the mouse retina. The data are **solid**, which provides **important** insights into how cell-adhesion molecules contribute to retinal organization. However, further investigations are needed to clarify the mechanisms underlying the cellular disorganization phenotype in the retina and axonal projection to the brain.

**\*For correspondence:**
Xin.Duan@ucsf.edu

[†]These authors contributed equally to this work

**Abstract** Neurons use cell-adhesion molecules (CAMs) to interact with other neurons and the extracellular environment: the combination of CAMs specifies migration patterns, neuronal morphologies, and synaptic connections across diverse neuron types. Yet little is known regarding the intracellular signaling cascade mediating the CAM recognitions at the cell surface across different neuron types. Using mouse genetics and viral labeling, we investigated the neural developmental role of Afadin (Mandai et al., 1997; Takai and Nakanishi, 2003; Takahashi et al., 1999), a cytosolic adapter protein that connects multiple CAM families to intracellular F-actin. We introduced the conditional Afadin mouse mutant (Beaudoin et al., 2012) to an embryonic retinal Cre, *Six3^Cre* (Oliver et al., 1995; Liu and Cvekl, 2017; Diacou et al., 2018). We reported that the mouse mutants lead to the scrambled retinal neuron distribution, including bipolar cells (BCs), amacrine cells (ACs), and retinal ganglion cells (RGCs), across three cellular layers of the retina. This scrambled pattern was first reported here at neuron-type resolution. Importantly, the mutants do not display deficits for BCs, ACs, or RGCs in terms of neural fate specifications or survival. Additionally, the displayed RGC types still maintain synaptic partners with putative AC types, indicating that other molecular determinants instruct synaptic choices independent of Afadin. Lastly, there is a significant decline in visual function and mis-targeting of RGC axons to incorrect zones of the superior colliculus, one of the major retinorecipient areas. Collectively, our study uncovers a unique cellular role of Afadin in sorting retinal neuron types into proper cellular layers as the structural basis for orderly visual processing.

## Introduction

The central nervous system (CNS) consists of complex circuits that are established during development and fine-tuned through both activity-dependent and apoptotic mechanisms. However, how neurons form these complex circuits—and, more specifically, how cell surface molecules promote correct neuronal migration and synapse formation—is not fully resolved. Cell-adhesion molecules (CAMs) have been shown to be key mediators of CNS lamination and synaptogenesis. Work from ours and others focused on using the inner retina of the mouse as a developmental system to understand mechanisms regulating these developmental questions (*Graham and Duan, 2021*; *Lefebvre et al., 2015*; *Sun et al., 2013*; *Matsuoka et al., 2011*; *Hoon et al., 2014*). Specifically, our past studies showed that type II cadherin (Cdhs), in combination, plays key roles in establishing appropriate synapses between retinal ganglion cells (RGCs) and BCs, as well as RGCs with amacrine cells (ACs) (*Duan et al., 2014*; *Duan et al., 2018*). Among the molecular machinery that composes the adherens junction (AJ) complex, β-catenin has been shown to be important in neuronal laminar organization, leading to embryonic deficits and major retinal neuron loss (*Fu et al., 2006*).

While we assume that all intracellular components of cell-surface adhesion complexes are critical for retinal neuron survival and patterning, not all components of the AJ complex equally regulate the same aspects of these developmental programs. Afadin is a cytosolic adaptor protein that links Nectin, a $Ca^{2+}$-independent immunoglobulin-like CAM, to F-actin microfilaments in the cytoskeleton (*Mandai et al., 1997*; *Takai and Nakanishi, 2003*; *Takahashi et al., 1999*; *Takai et al., 2008*). Afadin recruits cadherins to AJs mediated by Nectin, p120-catenin, and α-catenin (*Takai and Nakanishi, 2003*; *Takahashi et al., 1999*). While Afadin has multiple direct interactions with AJ proteins, it is not a core part of the cadherin–catenin complex (*Sawyer et al., 2009*).

Past genetic studies in the mouse CNS showed that loss of Afadin in both the hippocampal and cortical regions of the mouse brain leads to a decrease in dendritic spine density and number of synapses, with variable effects on dendritic arborization (*Beaudoin et al., 2012*). Additionally, Afadin plays an important role in cortical lamination: deletion of Afadin in the mouse telencephalon leads to cellular mislocalization and a resultant double-cortex (*Yamamoto et al., 2015*; *Gil-Sanz et al., 2014*). Notably, the *drosophila* homologue of Afadin is called *Canoe* (*Yu and Zallen, 2020*; *Mandai et al., 2013*), where the mutant phenotypes in the ommatidial eye were likely closely tied to the disruption of cellular junctions or synaptic complex, though given the broad role of Afadin (Canoe), they may also be due to other cell-surface signaling pathways (*Matsuo et al., 1999*).

The mouse neural retina offers a laminarly organized structure and well-characterized cellular composition across three cellular layers. During development, retinal progenitor cells span the retinal neuroepithelium via basal and apical processes, proliferate via asymmetric and symmetric divisions at the ventricular surface, and differentiate into six neuronal types via both transcriptional regulatory networks and environmental cues—these include rod and cone photoreceptors (PRs), horizontal cells (HCs), BCs, ACs, RGCs, and one glial cell type, Müller glia (MGs) (*Turner and Cepko, 1987*; *Cepko, 2014*; *Livesey and Cepko, 2001*; *Yan et al., 2020*). Thus, the distinct locations and temporal order offer a clear system to examine the roles of multifaceted molecules in every step of development, such as that for Afadin. By restricting the roles of Afadin into restricted RGC subsets or AC subsets, our recent study linked Afadin to the combinatorial Cdh complex that enables the selective RGC-AC synaptic choice (*Duan et al., 2018*). Yet it is unknown what role Afadin plays in neuronal migration, neuronal layer sorting, and brain target selection. Here, we utilized a developmental neural retina-specific Cre driver (*Six3^{Cre}*) (*Oliver et al., 1995*; *Liu and Cvekl, 2017*; *Diacou et al., 2018*) to generate a conditional Afadin mutant (*Six3^{Cre}*; *Afadin^{F/F}*). This conditional mutant allows us to characterize the role of Afadin in early development. Here, we report that the Afadin mutant significantly alters retinal neuronal migration and neuronal layer sorting, though it has little effect on cellular differentiation within the inner retina.

## Results

### Early Afadin conditional mutant scrambles retinal neuron layer organization

We utilized a murine conditional knockout model in which exon 2 of Afadin was flanked by LoxP sites (*Figure 1U*; *Beaudoin et al., 2012*). Cre-mediated recombination results in the excision of

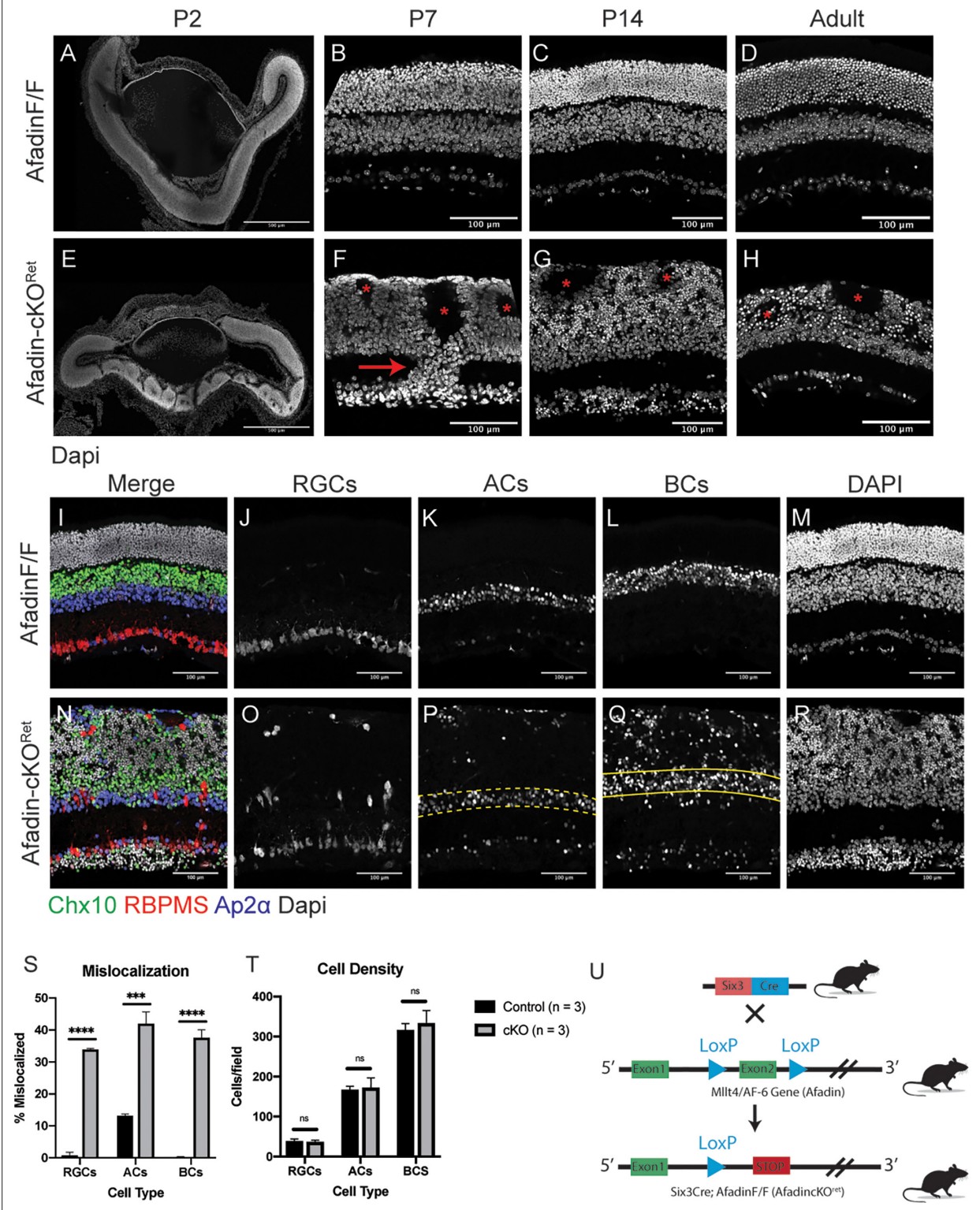

**Figure 1.** Retina-specific Afadin conditional mutants disrupt the cellular layer organization. (**A–H**) Postnatal time course of mouse retinal cryosections in Afadin control (*Afadin^F/F^*) and Afadin knockout (AfadincKO). The *Six3^Cre^*-Afadin knockout (**E–H**) displays disruption of the tri-neuronal layer organization. As a result, it leads to a fused singular outer layer (fused INL/ONL) compared to control retinae (**A–D**), which contain three distinct nuclear laminae separated by two plexiform layers. Rosettes (**F–H**, asterisks) in the fused INL/ONL are visible from as early as P2 to adulthood and are devoid of cell bodies and primarily contain neurites (see *Figure 2G, K and L*) (**F–H**). The IPL is retained in AfadincKO but contains columnar-like structures of displaced neurons (F, arrow). In adult AfadincKO mice (**H**), there is significant shrinkage of the fused INL/ONL (see *Figure 4E–H*). Scale bars

*Figure 1 continued on next page*

*Figure 1 continued*

(**A–H**): 100 μm. (**I–T**) Afadin conditional knockout results in the mislocalization of major retinal cell types. In cross-section view (**I–R**), the control retina displays stereotypical lamination of three major cell types: bipolar cells (Chx10), retinal ganglion cells (RBPMS), and amacrine cells (AP2a) (**I–M**). In control retinae, RGCs and BCs stayed in the GCL and INL strictly, with very little displacement. ACs have about 13.2 ± 0.4% displacement (**S**). In contrast, AfadincKO showed aberrant localization of three major cell types (**N–R**). RGCs, ACs, and BCs display 33.9 ± 0.4%, 42.0 ± 3.7%, and 37.6 ± 2.4%, respectively (**S**). Across three replicates, there was no significant difference between cell counts across the three cell types (**T**). Unpaired two-sided Student's *t*-tests; n.s., not significant; ****p<0.0001; ***p<0.001. Data presented as mean percentage mislocalized ± SEM. Mislocalization and cell density quantification were obtained from P14 mice. n=3 mice in each condition. Scale bars (**I–T**): 100 μm. (**U**) Generation of AfadincKO mice. *Six3^Cre* transgenic mouse crossed with Afadin conditional knockout mouse. Exon 2 is flanked by LoxP sites, enabling Cre-mediated deletion, resulting in a frameshift and premature stop codon.

The online version of this article includes the following source data for figure 1:

**Source data 1.** Cell type quantifications.

exon 2, resulting in a frameshift mutation and a premature stop codon. These conditional alleles were crossed with *Six3^Cre* transgenic mice to mediate gene deletion, particularly within the developing retinal neuroepithelium starting at E9 (*Oliver et al., 1995*; *Liu and Cvekl, 2017*; *Diacou et al., 2018*). Upon conditional knockout, Afadin mutants (hereafter referred to as AfadincKO) displayed aberrant lamination patterning at four different postnatal time points examined: P2, P7, P14, and P60 (*Figure 1E–H*). At P2, control retinas (*Afadin^F/F*) exist as a singularly laminated piece of tissue resulting from the proliferation and early differentiation of early-born retinal neurons (*Figure 1A*). In contrast, AfadincKO, the central retina is disrupted and contains rosette-like structures with regions devoid of cells at P2 (*Figure 1E*). At P7, the inner and outer plexiform layers are evident in control mice (*Figure 1B*) but notably disrupted in AfadincKO, with the OPL omitted and instead existing as a singular fused nuclear layer (fused INL/ONL) (*Figure 1F*). P7 and older mutants also show columns of neurons (*Figure 1F*, arrow; *Figure 2F*) spanning the inner plexiform layer (IPL) and rosettes in the fused INL/ONL (*Figure 1F*, asterisks). By P14, most, if not all, retinal neuron types and Müller glial cells have been established (*Turner and Cepko, 1987*; *Cepko, 2014*). To determine whether loss of Afadin affects retinal neuron densities or cell fate differentiation in addition to cellular layer organization, we examined mouse retinas and quantified neuronal subtypes in controls and mutants at P14 using well-established molecular markers: RBPMS to label RGCs, Chx10 to label BCs, and AP2α to label ACs. We found that the densities of the three major inner retinal neuronal types did not differ significantly between controls and mutants (*Figure 1T*), indicating retinal neurons in AfadincKO differentiate and proliferate via expected proportions, and Afadin is likely not involved in fate determination or cell survival regulations. When we used P14 mice to quantify the mislocalization of retinal types across three cellular layers, RGCs, ACs, and BCs were significantly scrambled across three cellular layers, compared to control (*Figure 1S*). Collectively, these results revealed the roles of Afadin in sorting inner retinal neurons into proper layers, likely integrating the positioning cues from the environment when specifying the layers.

## Mis-localized neurons in AfadincKO form an ectopic inner plexiform layer

In addition to scrambled neuronal locations in the wrong cellular layers, we also observed unusual outer layer 'rosette' structures and IPL bridging columns in AfadincKO that were evident as early as P2. To better characterize these substructures in our model, we utilized wholemount retina sections to obtain an *en face* view of the IPL and fused INL/ONL (*Figure 2E–H*). Interestingly, these rosettes retained some degree of canonical neuronal patterning (*Figure 2—figure supplement 1A–F*). They comprised radially arranged rod BCs projecting inwards towards ACs and RGCs (*Figure 2I and J*). The subtype identities of RGCs within the clusters were diverse, including Osteopontin (Spp1)[+] αRGCs (*Zhao et al., 2023*; *Duan et al., 2015*), Melanopsin (Opn4)[+] intrinsically photosensitive RGCs (ipRGCs) (*Provencio et al., 2000*; *Provencio et al., 1998*), and Cartpt[+] ON-OFF direction-selective ganglion cells (ooDSGCs) (*Kay et al., 2011*). In addition, dendrites of Carpt[+] ooDSGCs are projected centrally. They colocalized with starburst amacrine cell (SAC) dendrites (*Figure 2J and K* and *Figure 2—figure supplement 1D*), reminiscent of traditional synaptic pairings at the IPL among these cell types (*Wei and Feller, 2011*; *Wei et al., 2011*). We next quantified the number of 'rosettes', which numbered approximately 160±13 (*Figure 2—figure supplement 2A and B*) across four AfadincKO retinas in

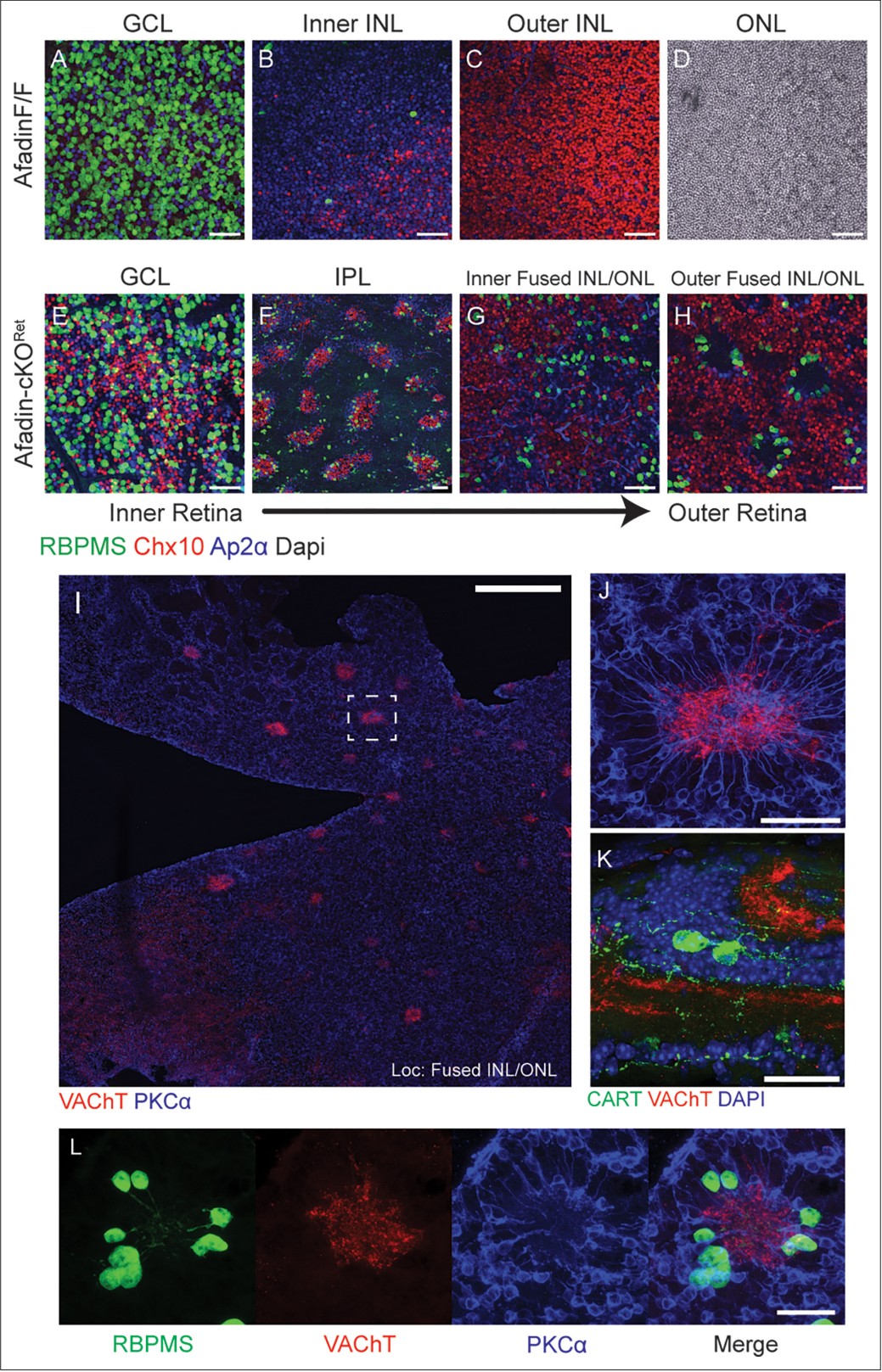

**Figure 2.** Synaptic rosettes persist despite lateral displacement of cell types. (**A–H**) Wholemount retinal cross-sections display lateral displacement of major cell types in AfadincKO. In a wholemount section view, AfadincKO (**E–H**) displays lateral displacement of cell types within unexpected laminae, which are absent in control (**A–D**). Notably, BCs (Chx10) are seen in the GCL (**E**), and RGCs are visible in the fused INL/ONL (**G, H**). ACs are

*Figure 2 continued on next page*

*Figure 2 continued*

found throughout all laminae in AfadincKO. The IPL in the AfadincKO contains regularly interspaced clusters of RGCs, BCs, and ACs, which form vertical bridging columns (**F**); IPL for the control retina is not shown. The rosettes are visible, with the neurites of RBPMS +RGCs projecting inward toward the rosette center (**H**). Scale bars (**A–H**): 50 µm. (**I–L**) Rosettes in the fused INL/ONL have characteristics of an ectopic IPL. Wholemount section (**I**) of an outer region of the fused INL/ONL showing starburst amacrine cell (SAC) processes labeled by VAChT forming a central rosette structure upon which BC processes colocalize. In a 60x magnification of the dashed region in (**I**), the spoke-like processes of rod bipolar cells stained with PKCα are visible (**J**). RGC dendrites also co-cluster in the rosette structure projecting centrally (**L**, RBPMS). A subset of these RGCs is Cartpt-positive, which labels ooDSGCs, indicating an ectopic IPL circuit composed of DSGCs is retained at the histological level (**K**). Scale bar (**I**): 300 µm; Scale bar (**J**): 50 µm; Scale bars (**K, L**): 30 µm.

The online version of this article includes the following source data and figure supplement(s) for figure 2:

**Figure supplement 1.** Canonical synaptic pairs persist in AfadincKO despite mislocalization (related to *Figure 2*).

**Figure supplement 2.** Representative AfadincKO wholemount.

**Figure supplement 2—source data 1.** Rosette count.

adults. This suggested that rosette formation was likely not a random occurrence but was likely a result of compensatory mechanisms of retinal lamination independent of Afadin and its CAM partners.

## Axon pathfinding to central visual targets remains intact in AfadincKO

Prior work exploring the role of Afadin and N-cadherin in the mouse dorsal telencephalon revealed that loss of either led to severe defects in axonal pathfinding, in addition to neuronal mislocalization and increased progenitor cell proliferation (*Yamamoto et al., 2015*; *Gil-Sanz et al., 2014*; *Rakotoma-monjy et al., 2017*). We asked whether loss of Afadin in the retina would affect RGC projections to the superior colliculus (SC), one of the primary retinorecipient areas in the mouse receiving input from more than 85% of RGCs (*Cang et al., 2018*; *Matcham et al., 2024*; *Tsai et al., 2022*). To delineate the contributions from the right and left eyes, we administered intravitreal injections with CTB-488 and CTB-555 to label projection neurons, that is, RGCs from the eyes: these dyes are subsequently transported via RGC axons to the SC (*Figure 3A*). Interestingly, though RGCs did project to the SC, ipsilateral/contralateral segregation was disrupted. While most (>97%*) projecting axons should cross at the midline in mice and project contralaterally, we noted an unusually high number of aberrant ipsilateral projections in AfadincKO (*Figure 3A*). Using an AAV(Retro)-mediated axonal projection-based retrograde labeling for RGC labeling, we delivered AAV-Retro TdTomato into the brain targets. We observed retrogradely labeled RGC distributions inside the retina (*Figure 3B*). To our surprise, the retrograde labeling also labeled RGCs displaced into the fused INL/ONL (*Figure 3C–E*). Approximately 42.2 ± 8.8% of the TdTomato labeled RGCs were mislocalized in AfadincKO (*Figure 3F*), while only a few RGCs were displaced into the INL in the control conditions (*Figure 3C and F*). Altogether, these results suggest that Afadin loss leads to axonal pathfinding deficits; on the other hand, the scrambled RGCs still grow their axons onto the central targets, including the SC.

## Photoreceptor loss in AfadincKO disrupts visual function

Associated with Afadin mutants in the inner retina, we inquired about the functional changes associated with such drastic anatomical changes. We observed a progressive PR loss within the same mutants. Thus, the mutants in their current form prevented us from further inquiring about functional changes associated with the inner retina physiological functions and instead encouraged examination of how Afadin loss affected PR lamination and survival. In adult mice at P60 (*Figure 4A and B*), we observed a close-to-complete loss of central PRs; there was noted peripheral sparing, as *Six3^Cre* drives primarily central retinal neuroepithelium during development. Using Recoverin, which primarily labels rods and a small subset of BCs, we observed a significant loss of Recoverin⁺ rods in AfadincKO mice (*Figure 4C–J*). The mechanisms leading to the loss of PRs are currently unknown; however, the loss is likely due to the postnatal disruption of the organization and connectivity. Yet, the same mutants caused no major losses of BCs, ACs, and RGCs (*Figure 1T*). Using the same set of mutants at P60, we sought to obtain a comprehensive measurement of visual function in AfadincKO mice. Given the degree of disorganization and PR loss, we noted significantly reduced scotopic and photopic ERG responses (*Figure 4M*), with flattened a and b waves in scotopic conditions (*Figure 4K and L*).

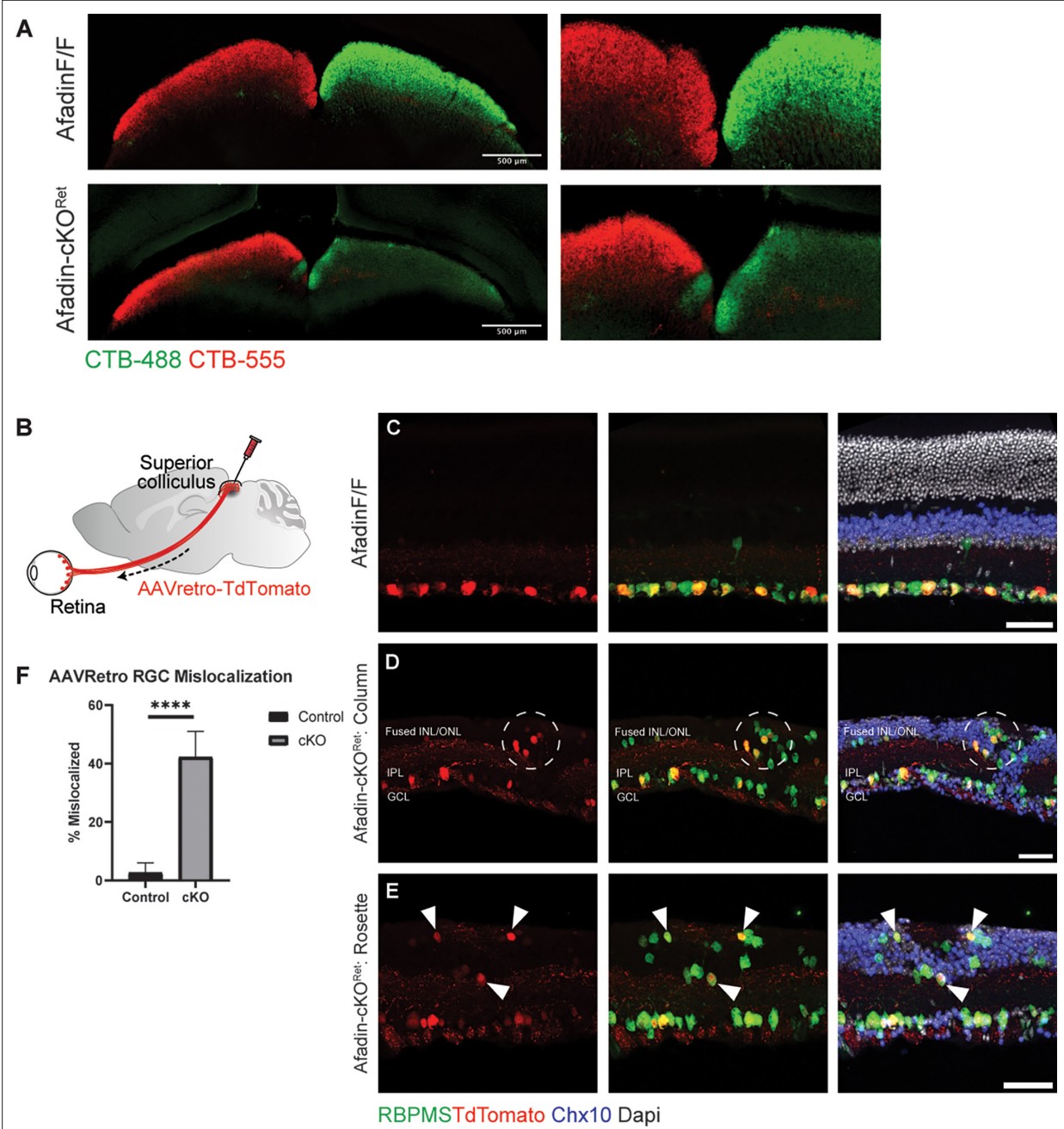

CTB-488 CTB-555

RBPMS TdTomato Chx10 Dapi

**Figure 3.** A significant fraction of displaced RGCs project to the central targets in the SC. (**A**) SC sections labeled with bilateral retina injections. CTB-488 and CTB-555 dye were injected into the left and right eye, respectively, of both *Afadin*F/F and AfadincKO mice. An ectopic CTB-488 patch was found close to the midline of the left SC. Scale bar: 500 um. (**B**) Illustration showing stereotaxic protocol. AAV (Retro)-TdTomato was injected unilaterally into the right SC of adult mice. The retrograde virus was uptaken by RGC terminals in the SC and selectively labeled RGC somata in the retina. (**C–E**) Displaced RGCs in AfadincKO send axons to the SC. Retinal cross-sections of *Afadin*F/F mice show retrograde-AAV labeled RGCs restricted to the GCL layer (**C**). In AfadincKO sections, RGCs co-labeled by RBPMS and TdTomato are mislocalized into the fused INL/ONL (dashed circle) and are close to an IPL column (**D**). TdTomato-positive RGCs (arrows) are also mislocalized to a rosette structure in the fused INL/ONL (**E**). Scale bar: 50 um. (**F**) Quantifications of AAVretro-labeled RGC somata. 2.7 ± 2.4% of RGCs co-labeled with RBPMS and anti-TdTomato in control mice displayed mislocalization beyond GCL, versus 42.2 ± 8.8% of RGCs in AfadincKO. Unpaired two-sided Student's *t*-tests; ****p<0.0001.

The online version of this article includes the following source data for figure 3:

**Source data 1.** AAV retrograde data.

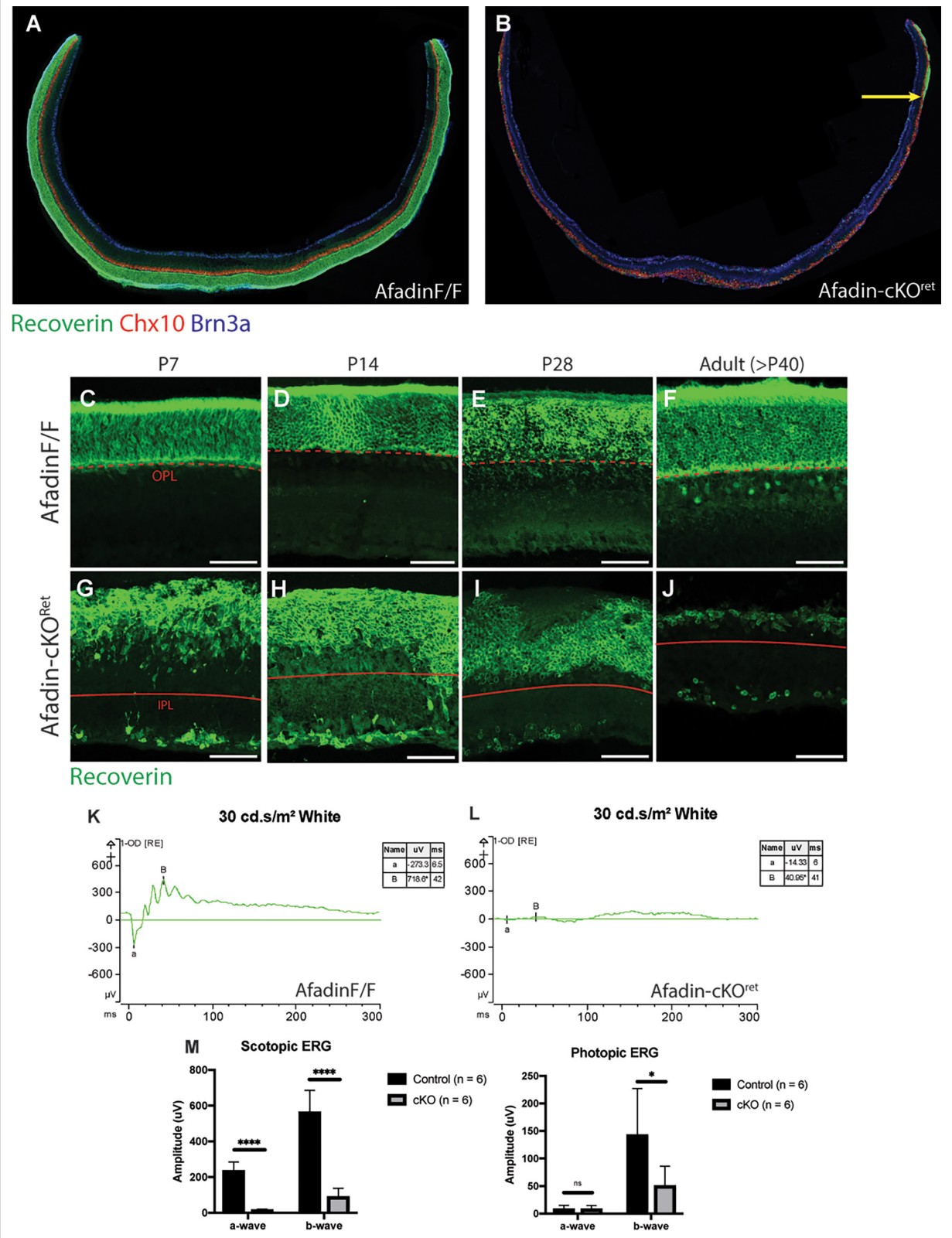

**Figure 4.** Afadin mutants lose photoreceptor-mediated visual function in adults. (**A, B**) Retinal cryosection displaying thinning and rod photoreceptor loss. Retinal cryosection taken from the central retina of AfadincKO (**B**) displays loss of recoverin-positive rod photoreceptors; the photoreceptor layer is maintained in control (**A**). Yellow arrow: displays remnant recoverin-positive patch in the peripheral retina. Scale bars: 500 um. (**C–J**) Postnatal time course of AfadincKO displaying progressive photoreceptor loss. Recoverin-positive photoreceptors display aberrant lamination in AfadincKO (**G–J**) but

*Figure 4 continued on next page*

*Figure 4 continued*

not in *Afadin^{F/F}* (**C–F**). By early adulthood, AfadincKO have a near-complete loss of photoreceptors (**J**). Dashed red lines indicate the transition from ONL to OPL. Solid red lines indicate the transition from fused INL/ONL to IPL. Scale bars: 50 um. (**K–M**) Representative ERG traces shown for one right eye of both *Afadin^{F/F}* (**K**) and AfadincKO (**L**) mouse when shown a dark-adapted intensity program of 30 cd.s/m² white stimulus. In the control mouse, the a-wave amplitude was –273.3 uV, and the b-wave amplitude was 718.6 uV. In the mutant, the a-wave amplitude was –14.33 uV, and the b-wave amplitude was 40.95 uV. The ERG responses were quantified in (**M**). Average a-wave and b-wave responses for *Afadin^{F/F}* and AfadincKO mice after dark-adapting overnight. Under scotopic conditions using 30 cd·s/m² flash of white light, the average a-wave response was 235.6±49.0 uV for control and 17.0±4.3 uV for AfadincKO mice; the average b-wave response was 564.4±121.2 uV for control and 89.4±48.3 uV for AfadincKO mice. Under photopic conditions using 10 cd·s/m² flash of white light, the average a-wave response was 8.2±6.7 uV for control and 8.3±6.3 uV for AfadincKO mice; the average b-wave response was 142.7±84.2 uV for control and 50.4±35.4 uV for AfadincKO mice. Unpaired two-sided Student's *t*-tests; ns, not significant; ****p<0.0001; *p<0.1. Data presented as mean wave response (uV) ± SEM between right and left eyes across 6 different mice. ERG quantification was obtained from adult mice. n=6 mice per condition.

The online version of this article includes the following source data for figure 4:

**Source data 1.** ERG quantification.

**Source data 2.** Raw electrophysiology ERG data exported from Diagnosys machine.

Altogether, these results indicate that loss of Afadin affects PR stability, leading to significant visual deficits.

## Discussion

Herein, we demonstrate that developmental loss of Afadin significantly affects normal retinal development. We observed profound disorganization of retinal neuronal layers, including RGCs, ACs, and BCs. Interestingly, we also noted the appearance of rosette-like structures containing SACs and radially directed RGC dendrites, suggesting some retention of cellular organization and perhaps indicative of other compensatory mechanisms that regulate synaptogenesis. This disorganization also affects RGC axonal projections onto the SC. The roles of Afadin in CNS development were examined in other regions of the CNS: In the hippocampus, Afadin loss leads to mislocalization of CA1 and CA3 pyramidal cells; additionally, the spine density of CA1 pyramidal cell neurons is reduced, with the authors postulating that that may be a consequence of reduced cadherin puncta density (*Beaudoin et al., 2012*; *Toyoshima et al., 2014*). Similarly, loss of Afadin in the dorsal telencephalon leads to both the dispersion of neural progenitor cells and an increase in their total numbers. Neuronal differentiation was not significantly affected, but cells were localized to inappropriate cortical layers (*Rakotomamonjy et al., 2017*). Interestingly, while we also noted retinal neuron mislocalization into the wrong layers, we did not observe major differences in cell numbers among the retinal subtypes we profiled. This may speak to the varied roles of Afadin or their constituent adherent complexes in different parts of the brain and spinal cord.

It is being increasingly appreciated that cell adhesion complexes play a role in mediating neuronal migration, cellular layer sorting, and synaptogenesis (*Yamagata and Sanes, 2008*; *Yamagata et al., 2003*). Our work utilized the developing retina to elucidate some of the partners that mediate these interactions. However, our current study raises several questions that remain to be explored: What other mechanisms drive synaptogenesis within the retina; additionally, what is the role of cell adhesion complexes in regulating axonal pathfinding? The rosette-like structures that retain elements of normal synaptic pairings suggest that synaptogenesis may be a layered process, largely driven by cell adhesion complexes but possibly fine-tuned by other extracellular or intracellular mechanisms, irrespective of activity. Indeed, considerable work has shown that gap junctions and the electrical synapses they facilitate are precursors to the eventual formation of chemical synapses between neuronal pairs. Moreover, through what mechanisms does the loss of Afadin disrupt RGC projections and pathfinding? In the spinal neuroepithelium, deletion of Afadin leads to miswiring of motor circuits, such that certain, typically ipsilaterally projecting neurons instead project bilaterally in the Afadin knockout, leading to loss of left-right limb (*Dewitz et al., 2018*; *Skarlatou et al., 2020*). They propose that this is due to a duplication of the central canal that alters midline signaling within the spinal cord. Whether similar structural abnormalities exist in the SC and to what extent midline signaling is compromised remains to be examined.

## Materials and methods

### Animal experiments

Mice were maintained under regular housing conditions with standard access to food and drink in a pathogen-free facility. Male and female mice were used in roughly equal numbers; no sexual dimorphisms were observed. Animals with noticeable health problems or abnormalities were not used. All ages and numbers were documented. PCR of tail biopsy PCR determined genotypes.

The following mouse lines were used:

1. *Six3*$^{Cre}$ expresses Cre recombinase in all of the retina except its far periphery as previously described (*Lefebvre et al., 2015*; *Duan et al., 2018*).
2. *Afadin*$^{F/F}$ mice were generated by targeting the second exon of Afadin with flanking loxP sites as previously characterized in *Beaudoin et al., 2012*.

### Electroretinogram (ERG) recording

Six *Afadin*$^{F/F}$ and six AfadincKO mice were dark-adapted overnight, and ERG data was collected in dim red light. Mice were first anesthetized with a combination of ketamine/xylazine/acepromazine (70/10/2 mg/kg), and proparacaine eye drops were administered as local anesthesia. Pupils were then dilated with 1% Tropicamide. The mouse was placed on a heating pad (39°C) under a dim red light provided by the overhead lamp of the Diagnosys Celeris ERG apparatus (Diagnosys LLC). The light-guide electrodes were placed onto the corneas. For scotopic conditions, we used a white stimulus with 30 cd·s/m$^2$ luminance intensity. Signals were captured for 300 ms after each step to assess scotopic a- and b-wave function. Following the dark-adapted protocol, we used a photopic intensity ramp protocol to assess function in a light-adapted state. For photopic conditions, we used a white stimulus with 10 cd·s/m$^2$ luminance intensity. Signals were captured for 300ms after each step to assess scotopic a- and b-wave function. Following the recordings, each mouse was placed in its home cage on a heating pad (39°C) to aid recovery from anesthesia.

### Intravitreal Injection

Intravitreal injection protocol, as previously established in *Zhao et al., 2023*. Mice were first anesthetized with a combination of ketamine/xylazine/acepromazine (70/10/2 mg/kg). Then, CTB-488 (Invitrogen, C34775) and CTB-555 (Invitrogen, C34776) dye was injected into the vitreous chamber of the right and left eye, respectively, with a fine glass pipette (Sutter Instrument Company). The toxin was allowed to travel anterograde for 2 weeks before processing brain tissue.

### Stereotaxic injection into the superior colliculus for AAV-Retro

Stereotaxic injection protocol as previously established by *Tsai et al., 2022*. Mice were anesthetized with continuous 2% isoflurane/oxygen on a stereotaxic setup (Model 940, David Kopf Instruments). Meloxicam (5 mg/kg) was administered IP before the surgery and for two consecutive days after the surgery. AAV viruses were loaded into a pulled glass pipette connected with a syringe (Hamilton, 7634-O) by a dual ferrule adaptor (Hamilton, 55750–0). Injection speed and volume were controlled by a Microinjection Syringe pump (WPI, UMP3T-1). AAV (Retrograde, RG) -Cag-tdTomato (Addgene, 59462-AAVrg) was injected into the right and left SC of P60+adult *Afadin*$^{F/F}$ (control) and AfadincKO mice. Coordinates for superior colliculi injection: (3.9–4.2 mm posterior, 0.6–0.7 mm lateral to bregma, and 1.4–1.0 mm below the skull). Volume: 600 nl for a saturated SC injection. Three weeks later, the mice were humanely euthanized via transcardial perfusion. Brain and retinas were collected in 4% PFA for immunohistochemical analysis.

### Histology and image acquisition

Retina section histology as previously established (*Toma et al., 2024*). Retina wholemount protocols were previously described in *Duan et al., 2018*; *Duan et al., 2015*. A lethal overdose of anesthesia sacrificed the mice. The eyes were dissected and post-fixed with 4% PFA on ice for 1 h and rinsed with 1× PBS. Retinas were analyzed as cryosections and whole mounts. For frozen sections, tissues were immersed in 30% sucrose for 2 h, then frozen in OCT before sectioning in a cryostat (20 μm). For immunohistochemistry, sections were incubated in PBS with 3% donkey serum and 0.3% Triton X-100 for 1 h blocking, followed by primary antibodies overnight at 4°C. For wholemount retinas,

tissues were incubated with blocking buffer (5% normal donkey serum, 0.5% Triton X-100 in 1× PBS) overnight, followed by primary antibodies for 2–4 days at 4°C. Secondary antibodies were applied for 2 h at room temperature. Sections and wholemounts were washed with 1× PBS and mounted using Fluoromount-G Mounting Medium, with and without DAPI (Invitrogen). Confocal images were acquired using a Zeiss LSM900 (Carl Zeiss Microscopy).

## Wholemount sectioning

Retina section histology was described in (N. Tsai, M.R.L., X. D, manuscript in preparation). Eyes were collected and fixed in 4% PFA on ice for 30 min, dissected to remove the cornea and lens, and then placed back into fresh 4% PFA on ice for another 30 min. The sclera was peeled off from each retina, and four radial cuts were made. Retinas were then placed in 30% sucrose/PBS and kept at 4°C until the retinas equilibrated/sank. Retinas were mounted onto a 0.45 µm membrane filter (MF-Millipore, HABG01300) and stretched until flat. The retina and filter paper then underwent two cycles of drying and rewetting and 30% sucrose/PBS before drying for 5 min. The membrane filter was then trimmed to match the retina size. The retina and filter paper were then adhered onto a stage made through mounting a block of tissue freezing medium (EMS, 72592) by mounting a block of tissue freezing medium (EMS, 72592), which was formed earlier within an embedding mold (Polysciences) onto a cryostat chuck and sectioning to form a flat stage. The retina was then quickly embedded with a thin layer of tissue-freezing medium. The chuck was placed back onto a block of dry ice to solidify the tissue freezing tissue-freezing medium. The embedded retina block with chuck was then incubated at –80°C overnight. Following equilibration within the cryostat, the retina was cut flat by maintaining the orientation of the chuck with the cryostat blade while preparing the block and subsequent sectioning. Sections of 12 um thickness were collected onto superfrost plus slides (Fisherbrand, 12-550-1).

## Reagent and resource sharing

Requests for reagents and further inquiries may be directed to the corresponding author, Xin Duan ( xin.duan@ucsf.edu).

## Quantifications and statistical analysis

GraphPad Prism 9/10 was used to generate all graphs and complete all statistical analyses. Statistical significance definitions: n.s., not significant; *p<0.05; **p<0.01; ***p<0.001; ****p<0.0001. All data are presented as means ± SEM unless stated otherwise.

### Mislocalization and cell density quantification

Mislocalization and cell density images were analyzed using ImageJ software (NIH). In practice, every eighth section was systematically sampled during cryostat preparation, thus ensuring coverage of the entire visual field. The boundaries of the INL-IPL, IPL-GCL, and AC_layer-BC_layer were used as landmarks for mislocalization quantifications. Definitions of mislocalization: RGCs were considered mislocalized if present past the INL-IPL boundary (outer retina) or within the IPL columns; ACs were considered mislocalized if present in the GCL, IPL columns, or past the AC_layer-BC_layer (outer retina); BCs were considered mislocalized if present in the GCL, IPL columns, or within the AC_layer. Notably, the mislocalization was very robust and apparent to multiple co-authors. Five images were analyzed for each mouse (three mice for control and three mice for KO). An unpaired two-sided *t*-test was used to determine the statistical significance of the mislocalization difference between cell types across control and cKO conditions. Data presented as mean percentage mislocalized ± SEM.

### AAV-retro mislocalization

Definitions of mislocalization for AAV-retro: TdTomato-positive RGCs were considered mislocalized if present past the INL-IPL boundary (outer retina) or within the IPL columns. Notably, the mislocalization was very robust and apparent to multiple co-authors. An unpaired two-sided t-test was used to determine the significance between control and cKO conditions. Data presented as Mean percentage mislocalized ± SEM.

## Rosette count

Rosette counts were obtained manually using Cell Counter in ImageJ across four adult AfadincKO mice ranging in age from P40 to P70. Data presented as Mean rosette count ± SD.

## Reagents and resources

| Primary antibodies | | |
| --- | --- | --- |
| Antibodies | Source | Catalog number |
| Chicken anti-GFP | Abcam | ab13970 |
| Rabbit anti-RFP | Rockland | 600-401-379 |
| Rabbit anti-RBPMS | Proteintech | 15187-1-AP |
| Rabbit anti-Recoverin | Millipore | AB5585 |
| Rabbit anti-Cartpt | Phoenix Pharmaceuticals | H-003-62 |
| Goat anti-Chx10 | Santa Cruz Biotechnology | sc-21690 |
| Goat anti-VAChT | Promega | G4481 |
| Goat anti-Chat | Millipore | AB144P |
| Mouse anti-Brn3a | Millipore | mab1585 |
| Mouse anti-PKCα | Abcam | AB11723 |
| Rat anti-tdTomato | Kerafast | EST203 |
| Secondary antibodies | | |
| Antibody | Source | Identifier |
| Alexa Fluor 488 | Invitrogen | - |
| Alexa Fluor 568 | Invitrogen | - |
| Alexa Fluor 633 | Invitrogen | - |
| AAV vectors | | |
| AAVrg-Cag-TdTomato | Addgene | 59462-AAVrg |

## Acknowledgements

We thank E Dang and L Pena for their assistance in animal care. We also thank YM Kuo and SL Wang for their technical support. We acknowledge (NEI P30EY002162) for vision core support, RPB unrestricted fund to UCSF-Ophthalmology; from Bright Focus Foundation Glaucoma Research Fellowship to MZ; from NIH (R01EY030138) and RBP Stein Award to XD; Glaucoma Research Foundation (Catalyst for a Cure to ALT, YH, DSW, and XD).

## Additional information

### Competing interests

Xin Duan: Reviewing editor, eLife. The other authors declare that no competing interests exist.

### Funding

| Funder | Grant reference number | Author |
| --- | --- | --- |
| National Eye Institute | R01EY030138 | Xin Duan |
| BrightFocus Foundation | G2024005F | Mengya Zhao |

| Funder | Grant reference number | Author |
|---|---|---|
| Glaucoma Research Foundation | CFC3 | Anna La Torre<br>Yang Hu<br>Derek S Welsbie<br>Xin Duan |
| Research to Prevent Blindness | Stein Innovation Award | Xin Duan |

The funders had no role in study design, data collection and interpretation, or the decision to submit the work for publication.

### Author contributions

Matthew R Lum, Data curation, Formal analysis, Methodology, Writing – original draft; Sachin Patel, Data curation, Formal analysis, Writing – original draft, Writing – review and editing; Hannah K Graham, Conceptualization, Data curation, Formal analysis, Methodology; Mengya Zhao, Investigation, Methodology; Yujuan Yi, Investigation; Liang Li, Resources, Methodology; Melissa Yao, Anna La Torre, Luca Della Santina, Methodology; Ying Han, Derek S Welsbie, Supervision; Yang Hu, Supervision, Methodology; Xin Duan, Conceptualization, Supervision, Funding acquisition, Writing – original draft, Writing – review and editing

### Author ORCIDs

Matthew R Lum ⓘ https://orcid.org/0000-0001-6992-4365
Liang Li ⓘ https://orcid.org/0000-0002-6292-5995
Yang Hu ⓘ https://orcid.org/0000-0002-7980-1649
Xin Duan ⓘ https://orcid.org/0000-0001-5260-8972

### Ethics

All animal experiments were approved by the Institutional Animal Care (IACUC) at the University of California at San Francisco (UCSF, IACUC approval: AN200631). Mice were maintained under standard housing conditions with unrestricted access to food and water in a pathogen-free facility.

Reviewer #1 (Public review): https://doi.org/10.7554/eLife.105575.2.sa1
Reviewer #2 (Public review): https://doi.org/10.7554/eLife.105575.2.sa2
Author response https://doi.org/10.7554/eLife.105575.2.sa3

## Additional files

### Supplementary files

MDAR checklist

### Data availability

All immunohistochemistry and electrophysiology data presented here are linked to the figures in the publication. Raw immunohistochemistry data related to the manuscript are deposited at https://www.ebi.ac.uk/biostudies/bioimages/studies/S-BIAD2528. Additional requests can be directed to the corresponding author, Xin Duan (xin.duan@ucsf.edu).

The following dataset was generated:

| Author(s) | Year | Dataset title | Dataset URL | Database and Identifier |
|---|---|---|---|---|
| Lum M, Duan X | 2025 | Afadin Sorts Different Retinal Neuron Types into Accurate Cellular Layers | https://www.ebi.ac.uk/biostudies/bioimages/studies/S-BIAD2528 | BioImage Archive, S-BIAD2528 |

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
