## [Editor Report · eLife Assessment]

This study demonstrates the critical role of Afadin on the generation and maintenance of complex cellular layers in the mouse retina. The data are **solid**, which provides **important** insights into how cell-adhesion molecules contribute to retinal organization. However, further investigations are needed to clarify the mechanisms underlying the cellular disorganization phenotype in the retina and axonal projection to the brain.

---

## [Referee Report · Reviewer #1 (Public review)]

Summary:

In this study, the authors examined the role of Afadin, a key adaptor protein associated with cell-adhesion molecules, in retinal development. Using a conditional knockout mouse line (Six3-Cre; AfadinF/F), the authors successfully characterized a disorganized pattern of various neuron types in the mutant retinae. Despite these altered distributions, the retinal neurons maintained normal cell numbers and seemingly preserved some synaptic connections. Notably, tracing results indicated mistargeting of retinal ganglion cell (RGC) axon projections to the superior colliculus, and electroretinography (ERG) analyses suggested deficits in visual functions.

Strengths:

This compelling study provides solid evidence addressing the important question of how cell-adhesion molecules influence neuronal development. Compared to previous research conducted in other parts of the central nervous system (CNS), the clearly defined lamination of cell types in the retina serves as a unique model for studying the aberrant neuronal localizations caused by Afadin knockout. The data suggest that cell-cell interactions are critical for retinal cellular organization and proper axon pathfinding, while aspects of cell fate determination and synaptogenesis remain less understood. This work has broad implications not only for retinal studies but also for developmental biology and regenerative medicine.

Weaknesses:

While the phenotypes observed in the Afadin knockout (cKO) mice are intriguing, I would expect to see evidence confirming that Afadin is indeed knocked out in the retina through immunostaining. Specifically, is Afadin knocked out only in certain retinal regions and not others, as suggested by Figures 4A-B? Are Afadin levels different among distinct neuron types, which could mean that its knockout may have a more pronounced impact on certain cell types, such as rods compared to others?

The authors suggest that synapses may form between canonical synaptic partners, based on the proximity of their processes (Figure 2). However, more solid evidence is needed to verify these synapses through the use of synaptic marker staining or transsynaptic labeling before drawing further conclusions.

Although the Afadin cKO mice displayed dramatic phenotypes, additional experiments are necessary to clarify the details of this process. By manipulating Afadin levels in specific cell types or at different developmental time points, we could gain a better understanding of how Afadin regulates accurate retinal lamination and axonal projection.

---

## [Referee Report · Reviewer #2 (Public review)]

Summary:

This study by Lum and colleagues reports on the role of Afadin, a cytosolic adapter protein that organizes multiple cell adhesion molecule families, in the generation and maintenance of complex cellular layers in the mouse retina. They used a conditional deletion approach, removing Afadin in retinal progenitors, and allowing them to analyze broad effects on retinal neuron development.

The study presents high-quality and extensive characterization of the cellular phenotypes, supporting the main conclusions of the paper. They show that Afadin loss results in significant disorganization of the retinal cellular layers and the neuropil, producing rosettes and displacement of cells away from their resident layers. The major classes of neurons in the inner retina are affected, and some neurons are, remarkably, displaced to the other side of the inner plexiform layer. Nevertheless, they mostly target their synaptic partners, including the RGCs to distant retinorecipient targets in the brain. The main conclusions are as follows. Afadin is necessary for establishing and maintaining the retinal architecture. It is not necessary for the generation of the correct numbers/densities of retinal neuron subtypes. Moreover, Afadin loss preserves associations between known synaptic partners and preserves axonal targeting to retinorecipient layers. The consequences on photoreceptor viability and visual processing are also interesting, underscoring the essential function for maintaining retinal structure and function. Overall the main conclusions describing the consequences are supported by the results.

Strengths:

The study provides new knowledge on the requirement of Afadin in retinal development. The introduction and discussion effectively set up the rationale for this work, and place it in the context of previous studies of Afadin in other regions of the CNS.

The study presents high-quality and extensive characterizations of the cellular phenotypes resulting from Afadin loss. By analyzing various aspects of retinal organization - from cellular densities to axon targeting to brain - the study narrows down the role of a structure for promoting the establishment of the layers, or maintenance. The data are straightforward and convincing, and the interpretations are bounded by the data shown (though minor weakness re. survival). Another important finding is that the targeting of retinal neuron processes to synaptic partners, including retinorecipient targets in the brain, are intact.

The study is important as it establishes a focused requirement for Afadin to set up and preserve the overall cellular organizations within the retinal tissue. The demonstration that Afadin is needed for photoreceptor viability and overall visual function enhances impact by establishing its functional importance.

The manuscript is well well-written and presented. The images are attractive and compelling, and the figures are well organized.

Weaknesses:

(1) Expanding on the developmental mechanism is beyond the scope of the study, and would not add to the main conclusions. However, the manuscript would be improved by providing more clarity on the developmental emergence of the defects. The study left me questioning whether the rosettes and cell displacements occur during earlier stages of retina development, or are progressive. For instance, do the RGCs migrate and establish within the GCL correctly at first, and then are displaced with the progressive disorganization? Or are they disorganized and delaminate en route? Images of RGC staining at P0, or earlier during their migration, would be informative. Data in Figure 1 is limited to DAPI staining at P7. Figure 4 shows an image of rod photoreceptors at P7, with their displacement in the GCL layer (and not contained within a rosette). Are the progenitors mislocalized due to delamination?

A few additional thoughts on how these defects compare to other mutants with rosettes might give us more context for understanding the results.

(2) The manuscript reports that the densities of major inner retinal classes are unaffected. There are a few details missing for this point. How were the cell densities quantified (in terms of ROI size), and normalized? This information is lacking in the methods. There is a striking thickening of the GCL in the DAPI-labeled images shown in Figure 1. What are these cells?

---

## [Author Response]

**Public Reviews:**
Reviewer #1 (Public review):Summary:In this study, the authors examined the role of Afadin, a key adaptor protein associated with cell-adhesion molecules, in retinal development. Using a conditional knockout mouse line (Six3-Cre; AfadinF/F), the authors successfully characterized a disorganized pattern of various neuron types in the mutant retinae. Despite these altered distributions, the retinal neurons maintained normal cell numbers and seemingly preserved some synaptic connections. Notably, tracing results indicated mistargeting of retinal ganglion cell (RGC) axon projections to the superior colliculus, and electroretinography (ERG) analyses suggested deficits in visual functions.

Thank you for the summary and highlights of our study. We appreciate the input from Reviewer 1 and the Editor on this study, with focus on laminar choices, synaptic choices and axonal projections.

Strengths:This compelling study provides solid evidence addressing the important question of how cell-adhesion molecules influence neuronal development. Compared to previous research conducted in other parts of the central nervous system (CNS), the clearly defined lamination of cell types in the retina serves as a unique model for studying the aberrant neuronal localizations caused by Afadin knockout. The data suggest that cell-cell interactions are critical for retinal cellular organization and proper axon pathfinding, while aspects of cell fate determination and synaptogenesis remain less understood. This work has broad implications not only for retinal studies but also for developmental biology and regenerative medicine.Weaknesses:While the phenotypes observed in the Afadin knockout (cKO) mice are intriguing, I would expect to see evidence confirming that Afadin is indeed knocked out in the retina through immunostaining. Specifically, is Afadin knocked out only in certain retinal regions and not others, as suggested by Figures 4A-B? Are Afadin levels different among distinct neuron types, which could mean that its knockout may have a more pronounced impact on certain cell types, such as rods compared to others?The authors suggest that synapses may form between canonical synaptic partners, based on the proximity of their processes (Figure 2). However, more solid evidence is needed to verify these synapses through the use of synaptic marker staining or transsynaptic labeling before drawing further conclusions.Although the Afadin cKO mice displayed dramatic phenotypes, additional experiments are necessary to clarify the details of this process. By manipulating Afadin levels in specific cell types or at different developmental time points, we could gain a better understanding of how Afadin regulates accurate retinal lamination and axonal projection.

Regarding the antibody confirming the Knockout, we tested the commercially available antibody from Sigma but weren’t able to confirm its specificity. There was a homemade antibody from another Japan-based laboratory, but it was not available to share at the moment when the study was conducted. Nonetheless, the original allele was derived for hippocampal and cortical studies by Louis Reichardt’s Lab (UCSF), with verified efficacies of the KO allele.

Regarding phenotypical penetrance, this may likely come from the mosaicism of the clone and the symmetric cell division, leading to a rosette-like structure. At this moment, we reason that Afadin KO does NOT lead to direct neuronal loss, and the selective rod loss may derive from other issues, but we lack direct evidence to validate this point.

In regards to the specific neuronal types and synaptic pairs, we acknowledge the limitations of the current Figure 2 in linking the mutant phenotypes to circuit changes. However, the current genetic reagents (Six3Cre) are not compatible with neuron-type specific labeling of synaptic labeling – i.e., cell type-specific Cre and additional Cre-dependent AAV tools might be desired. To do so, we will need to initiate cell-type-specific breeding of transgenic markers such as Hb9GFP for ooDSGCs, or Chat-Cre, VGlut3-Cre for starburst amacrine cells, vG3 amacrine cells, followed by retinal physiology. These experiments take multi-allelic genetic crosses for a very low breeding yield (1/16 or 1/32 Mendelian ratio). These extensive genetic tests are beyond the scope of the current manuscript.

**Reviewer #2 (Public review):**
Summary:This study by Lum and colleagues reports on the role of Afadin, a cytosolic adapter protein that organizes multiple cell adhesion molecule families, in the generation and maintenance of complex cellular layers in the mouse retina. They used a conditional deletion approach, removing Afadin in retinal progenitors, and allowing them to analyze broad effects on retinal neuron development.The study presents high-quality and extensive characterization of the cellular phenotypes, supporting the main conclusions of the paper. They show that Afadin loss results in significant disorganization of the retinal cellular layers and the neuropil, producing rosettes and displacement of cells away from their resident layers. The major classes of neurons in the inner retina are affected, and some neurons are, remarkably, displaced to the other side of the inner plexiform layer. Nevertheless, they mostly target their synaptic partners, including the RGCs to distant retinorecipient targets in the brain. The main conclusions are as follows. Afadin is necessary for establishing and maintaining the retinal architecture. It is not necessary for the generation of the correct numbers/densities of retinal neuron subtypes. Moreover, Afadin loss preserves associations between known synaptic partners and preserves axonal targeting to retinorecipient layers. The consequences on photoreceptor viability and visual processing are also interesting, underscoring the essential function for maintaining retinal structure and function. Overall, the main conclusions describing the consequences are supported by the results.Strengths:The study provides new knowledge on the requirement of Afadin in retinal development. The introduction and discussion effectively set up the rationale for this work, and place it in the context of previous studies of Afadin in other regions of the CNS.The study presents high-quality and extensive characterizations of the cellular phenotypes resulting from Afadin loss. By analyzing various aspects of retinal organization - from cellular densities to axon targeting to brain - the study narrows down the role of a structure for promoting the establishment of the layers, or maintenance. The data are straightforward and convincing, and the interpretations are bounded by the data shown (though minor weakness re. survival). Another important finding is that the targeting of retinal neuron processes to synaptic partners, including retinorecipient targets in the brain, are intact.The study is important as it establishes a focused requirement for Afadin to set up and preserve the overall cellular organizations within the retinal tissue. The demonstration that Afadin is needed for photoreceptor viability and overall visual function enhances impact by establishing its functional importance.The manuscript is well well-written and presented. The images are attractive and compelling, and the figures are well organized.

Thank you for your high praise on the logic, data presentation, and significance of the current manuscript. We appreciate your comments on the novelty and impact of our study using retinal circuits as a model.

Weaknesses:(1) Expanding on the developmental mechanism is beyond the scope of the study, and would not add to the main conclusions. However, the manuscript would be improved by providing more clarity on the developmental emergence of the defects. The study left me questioning whether the rosettes and cell displacements occur during earlier stages of retina development, or are progressive. For instance, do the RGCs migrate and establish within the GCL correctly at first, and then are displaced with the progressive disorganization? Or are they disorganized and delaminate en route? Images of RGC staining at P0, or earlier during their migration, would be informative. Data in Figure 1 is limited to DAPI staining at P7. Figure 4 shows an image of rod photoreceptors at P7, with their displacement in the GCL layer (and not contained within a rosette). Are the progenitors mislocalized due to delamination? A few additional thoughts on how these defects compare to other mutants with rosettes might give us more context for understanding the results.

We chose P7 as our focus due to the lamination in controls. In the revised manuscript, we plan to include earlier time points, as suggested by the reviewer. The data in Figure 1 at P7 utilizes well-established cell type markers (RBPMS, Chx10, Ap2α) and is not limited only to DAPI. Additionally, we will revise the discussion section and place our mutant analyses in the context of other mutants with rosettes (beta-catenin, etc.) in the retina. Finally, we will address the comment on progenitor lamination by exploring earlier developmental time points.

(2) The manuscript reports that the densities of major inner retinal classes are unaffected. There are a few details missing for this point. How were the cell densities quantified (in terms of ROI size), and normalized? This information is lacking in the methods. There is a striking thickening of the GCL in the DAPI-labeled images shown in Figure 1. What are these cells?

We will revise the manuscript, particularly the methods section, to address these comments. Additionally, we will tackle ROI units and normalization. The cells in the thickened GCL were identified as displaced amacrine cells and bipolar cells.